# Exploration of Psychiatry Residents’ Attitudes toward Patients with Substance Use Disorder, Bipolar Disorder and Schizophrenia in Saudi Arabia

**DOI:** 10.3390/bs13080642

**Published:** 2023-08-01

**Authors:** Abdullah M. Alarifi, Najim Z. Alshahrani, Nawaf H. Albali, Khalid M. Aljalajel, Nourh M. Alotaibi, Anan A. Fallatah, Majd Rachid Zeitounie, Khalid A. Alghamdi, Maan A. Alsaaid, Ahmed Alshehri

**Affiliations:** 1Department of Public Health, College of Health Sciences, Saudi Electronic University, Riyadh 13316, Saudi Arabia; 2Department of Family and Community Medicine, Faculty of Medicine, University of Jeddah, Jeddah 23218, Saudi Arabia; 3Department of Epidemiology, Biostatistics and Public Health, College of Medicine, Alfaisal University, Riyadh 11533, Saudi Arabia; 4Mental Health Department, King Faisal Specialist Hospital & Research Center, Riyadh 23433, Saudi Arabia; 5Department of Psychiatry, Ministry of Health, Riyadh 12613, Saudi Arabia; 6Eradah Complex for Mental Health, Ministry of Health, Riyadh 12613, Saudi Arabia; 7College of Medicine, Riyadh Campus, Alfaisal University, Riyadh 11533, Saudi Arabia; 8Adult Mental Health Department, King Abdulaziz Medical City, National Guard, Riyadh 11426, Saudi Arabia

**Keywords:** attitude, substance use, bipolar disorder, schizophrenia, psychiatry resident, Saudi Arabia

## Abstract

Stigmatizing attitudes of psychiatry professionals toward patients with various mental disorders may negatively impact treatment-seeking behaviors. However, in Saudi Arabia, little is known about psychiatry residents’ attitudes toward individuals with a specific disease/disorder. Therefore, the purpose of this study was to assess psychiatry residents’ attitudes toward patients with substance use disorder (SUD), bipolar disorder and schizophrenia in Saudi Arabia. Data for this cross-sectional study were collected from psychiatry residents (N = 79) in Saudi Arabia with a structured questionnaire containing sociodemographic and attitude-related variables. The 11-item Medical Condition Regard Scale (MCRS) for individuals with three conditions was used to assess participants’ attitudes. A linear regression model was fitted to investigate the association. Based on the MCRS (on a scale of 11 to 66), participants′ mean attitude scores were 41.59 (SD: 8.09), 54.53 (SD: 5.90) and 54.20 (SD: 6.60) for SUD, bipolar disorder and schizophrenia, respectively. Adjusted regression analysis demonstrated that senior residents, an age ≥ 27 years and a high confidence level were significantly associated with psychiatry residents’ positive attitudes toward patients with the three conditions. Psychiatry residents’ attitude scores were relatively lower (i.e., negative attitudes) for patients with SUD than for those with bipolar disorder and schizophrenia. Future longitudinal studies are recommended to explore the factors behind psychiatry residents’ negative attitudes toward patients with addictive behaviors and mental illnesses.

## 1. Introduction

Substance use disorder (SUD) is a chronic medical condition marked by severe or troublesome drug use that impairs individuals’ physical and mental health, as well as social functioning [1]. It has a combination of cognitive, behavioral and physiological outcomes that show that the user is persistently using the drug despite its adverse effects [1]. SUD is a major public health issue worldwide, including in Saudi Arabia [2]. Since Saudi Arabia is an Islamic country, religious beliefs have a significant role in cultural norms and values. Although alcohol and narcotics are illegal to possess and use in Saudi Arabia due to both religious and legal prohibitions, some Saudis do so and most are between the ages of 12 and 22 years [2,3]. Addiction to drugs or alcohol may increase the risk of productivity loss, accidents and absenteeism from work [4,5]. SUD is a treatable condition, and healthcare practitioners (such as psychiatrists, physicians, etc.) could play a crucial role in identifying and managing people with SUD [6].

In Saudi Arabia, another key area of public health concern is mental illnesses such as depression, schizophrenia, bipolar disorder, etc. [7,8]. Bipolar disorder is a type of personality disorder that involves mood fluctuations with at least one manic episode and may also feature recurrent depressive periods [1,9]. On the other hand, schizophrenia is a severe and chronic psychiatric condition that features psychotic symptoms (i.e., one is out of touch with reality) [1,9]. According to the Global Burden of Diseases, Injuries and Risk Factors Study (GBD) 2017, the years lived with disability (YLDs) due to mental disorders, SUD, neoplasms and neurological disorders consistently increased over the periods of 1990–2010 and 2010–2017 in Saudi Arabia [10]. A thematic analysis found that the shame associated with schizophrenia affected the quality of life of people with schizophrenia in Saudi Arabia [11]. In Arab countries, bipolar disorder has unique characteristics, such as high rates of mania and consanguinity, which provide opportunities for a more focused approach to make specific contributions to the field [12].

Given the high disease burden and poor prognosis rates of these conditions, proper knowledge and attitudes toward psychiatric disorders are of the utmost importance irrespective of any population group, such as the general public or students, as well as health professionals, to reduce negative beliefs and stigma. For instance, a Saudi Arabian study showed that the general population had suboptimal awareness, some misconceptions and negative attitudes regarding bipolar disorder [8]. According to a review study, stigmatizing attitudes regarding mental health issues are common among nursing students in Saudi Arabia [13]. In another Saudi study, both physicians and patients reported inadequate knowledge and unfavorable attitudes toward psychiatry [14].

Addiction research may benefit from the idea of addiction resistance, which evaluates individual variance in sensitivity to the emergence of SUD for a specific degree of drug or alcohol exposure [15]. For example, despite heavy use, some people are resistant to alcohol use disorders [16]. Addiction resistance is a measure of the discrepancy between alcohol intake and alcohol use disorder symptoms, where some people consume more alcohol but show fewer alcohol use disorder symptoms [16]. Better verbal fluency and interference–resistance abilities were associated with a higher propensity to use drugs to satiate cravings in SUD patients [17]. Persistent maladaptive memories that maintain drug-seeking and extinction resistance are a feature of addiction [18]. When it comes to commonly used psychoactive substances, addiction resistance is predicted by family history, comorbidities, personality, norm adherence and emotional stability [15,16].

Resistant bipolar disorder is one of the key mental health issues associated with significant disability, functional impairments and high overall costs [19]. Treatment for bipolar disorder is probably the most challenging of all mental disorders because each phase necessitates a different approach [20]. According to the literature, depression, not mania, is the phase that presents the greatest difficulties [21]. If it remains subsyndromal, the existence of persistent symptoms increases the probability of relapse and disability and reduces the overall prognosis [22]. In the case of schizophrenia, patients with resistant schizophrenia have a higher level of dissociation (i.e., a loss of integrity between memories and perceptions of reality) than patients in clinical remission [23]. Schizophrenia is a challenging condition because of the complex picture of altered perception, behavior, metabolic features and functional interconnectedness [23]. This is the reason that a significant proportion of patients continue to be resistive and present serious personal, family and societal issues despite the ongoing improvements in numerous therapeutic approaches [23,24]. Collectively, these could be the main reasons for failure in therapy, long stays in psychiatric wards, comorbidities as well as associated personality traits, leading to the development of stigmatizing attitudes toward a certain condition.

There is evidence that stigmatizing and unfavorable attitudes make people with mental health problems more distressed psychologically and also deter them from engaging in care-seeking behaviors [25,26]. In particular, stigmatizing attitudes of psychiatric specialists and other medical personnel toward patients with mental illness may negatively impact diagnosis and treatment [27,28]. Psychiatrists have reportedly shown higher stigmatizing attitudes toward patients with psychotic disorders, SUD and personality disorders, compared to those with other medical and physiological ailments [25,27,29].

Psychiatry residents (psychiatry residency training is a 4-year program to prepare residents to serve as consultants in psychiatry) are those physicians who have specialized to provide care for patients with co-occurring mental illnesses and SUD [27,28]. According to a prior US study, psychiatry residents’ attitudes toward people with SUD diagnoses, both with and without schizophrenia, were more stigmatizing [30]. This study also demonstrated that senior residents had more negative attitudes toward patients with SUD [30]. Given this fact, measuring psychiatry residents’ attitudes toward people with a particular condition/disorder is important in understanding the level of stigmatizing attitudes that may affect case management. However, in Saudi Arabia, little is known about psychiatry residents’ attitudes toward individuals with a specific disease/disorder. Therefore, we designed this study to bridge this research gap. The purpose of our study was to assess psychiatry residents’ attitudes toward patients with SUD, bipolar disorder and schizophrenia in Saudi Arabia.

## 2. Materials and Methods

### 2.1. Study Design and Ethics

This was an observational cross-sectional study. All study procedures were carried out in accordance with the Declaration of Helsinki, and the study protocol was ethically evaluated, reviewed and approved by the Research Ethics Committee at the Security Forces Hospital Program in the Holy Capital (HAP-02-K-052). The approval number was ECM#0500-140822. Informed consent (electronic written signature) was obtained from the surveyed individuals. The objectives of the study were clearly stated in the consent form, and participants were assured of the confidentiality of their data and privacy of their personal information. The study was conducted over two months, from 15 January to 15 March of 2023. The overall flow diagram of the study design is depicted in Figure 1.

### 2.2. Participants and Study Procedures

This study was performed among all psychiatry residents in Saudi Arabia to assess their attitudes toward patients with different psychiatric conditions. It was estimated that 80 psychiatry residents were currently working in Saudi Arabia, and all of them were invited to participate in the research. Finally, almost all agreed to take part in this survey (N = 79), with a response rate of 99%.

An online platform (a Google survey link was sent via email and shared personally via a QR code in weekly residents’ meetings) was used to gather the survey data from the participants. An online data collection technique was used so that psychiatry residents could provide their responses at a convenient time (as they might have been busy with their professional commitments). The study team followed two systematic approaches to examine the study subject. Initially, we contacted the chief and director of the psychiatry residents and took psychiatry residents’ contact details, including email addresses. Finally, we sent them (the study participants) an email with a link and a QR code to a survey containing a consent form and a structured questionnaire. In the body of the email, instructions were given on how they could provide consent and participate in the study. In the case of delayed responses, they were given a respectful reminder by sending a follow-up email on a weekly basis.

### 2.3. Study Variables and Measurement 

The questionnaire consisted of 45 variables under four sub-sections: (i) socio-demographic and confident-related information (twelve variables), (ii) assessment of psychiatry residents’ attitudes toward individuals with SUD (11 variables), (iii) assessment of psychiatry residents’ attitudes toward individuals with bipolar disorder (11 variables) and (iv) assessment of psychiatry residents’ attitudes toward individuals with schizophrenia.

#### 2.3.1. Outcome Measurement

Psychiatry residents’ attitudes toward patients with SUD, bipolar disorder and schizophrenia were the three dependent variables in this study. To assess the psychiatry residents’ attitudes regarding a given condition, we used the valid and reliable 11-item Medical Condition Regard Scale (MCRS) [31]. This scale measures the extent to which physicians find a patient with a given medical condition to be pleasurable, treatable and worthy of medical resources [31] This measure is considered as a proxy for “attitudes” [30]. The MCRS uses a six-point Likert scale that extends from strongly disagree to strongly agree. The scores range from 1 (strongly disagree) to 6 (strongly agree) for all of the items, except for item numbers 1, 2, 4, 9 and 11. Reverse scoring is employed for these five items (i.e., strongly agree = 1 to strongly disagree = 6). The overall score ranges from 11 to 66, with a higher score indicating a positive attitude (i.e., the least stigmatizing attitude) [30,32]. Previous epidemiological studies have used this scale in a variety of settings [30,32,33]. In the present study, a good level of internal consistency of this scale for SUD (Cronbach’s alpha = 0.772), bipolar disorder (Cronbach’s alpha = 0.725) and schizophrenia (Cronbach’s alpha = 0.793) was observed.

#### 2.3.2. Explanatory Variables

Socio-demographic information such as gender (male or female), age, marital status (married or single), level of residency (junior or senior), smoking status (yes or no), completion of addiction rotation (yes or no) and management of a number of cases in terms of SUD, bipolar disorder and schizophrenia (≥10 cases or <10 cases) was included as explanatory variables. Participants’ age was captured as a continuous scale and subsequently categorized into two groups (<27 years vs. ≥27 years) based on the median age of the sample. Residents were divided into two groups, representing junior and senior residents with postgraduate year 1 to 2 vs. 3 to 4, respectively. In addition, participants’ confidence levels in managing the SUD, bipolar and schizophrenia patients were assessed by the following question: “How confident are you in managing the following cases?” (not at all confident, somewhat confident or very confident).

### 2.4. Statistical Analysis

Both enumerative (such as frequency, percentage, mean and standard deviation) and analytical statistics were performed when analyzing the study data. The Shapiro–Wilk test was performed to check the distribution of the attitude scores (three dependent variables) and found that all attitude scores were normally distributed (i.e., attitude scores for SUD (W = 0.976, *p* = 0.143) and bipolar disorder (W = 0.979, *p* = 0.217)) except the attitude scores for schizophrenia (W = 0.962, *p* = 0.019). If the distribution of the attitude scores was normal, then a parametric test (such as independent-sample *t*-test or one-way ANOVA) was used to compare attitude scores across the different independent variables; otherwise, an alternative test was used (i.e., Mann–Whitney U test or Kruskal–Wallis test). Finally, three separate multiple linear models (Models 1 to 3) were used to identify the factors associated with attitudes toward patients with SUD, bipolar disorder and schizophrenia. All background variables, including the confidence level in managing the respective cases, were included in the adjusted regression model. All criteria related to linear regression were tested after fitting the model. The variance inflation factor (VIF) and tolerance were used to check for multicollinearity among the independent variables. A mean VIF of less than 10 was considered acceptable, according to earlier studies [34,35]. Data were represented as regression coefficients (β) with a 95% confidence interval (CI). A *p*-value of 0.05 was taken as a threshold for statistical significance. All analyses were performed using STATA (BE version 17.0, StataCorp, College Station, TX, USA) and SPSS (IBM version 23.0, Armonk, NY, USA).

## 3. Results

### 3.1. Sample Characteristics

The median age of the participants was 27.0 years, and more than half of the participants were male (53.2%). Over half of the participants (53.2%) were senior residents. The majority of respondents reported confidence (somewhat confident or very confident) in managing patients with SUD (89.9%), bipolar disorder (93.7%) and schizophrenia (92.4%) (Table 1).

### 3.2. Attitudes toward Patients with Different Conditions and Their Associated Factors

Based on the MCRS, the participants’ mean attitude scores were 41.59 (SD: 8.09), 54.53 (SD: 5.90) and 54.20 (SD: 6.60) for SUD, bipolar disorder and schizophrenia, respectively (on a scale of 11 to 66). Psychiatry residents’ attitude scores for SUD, bipolar disorder and schizophrenia varied significantly by age and residency level (*p* < 0.05). Participants’ attitude scores for bipolar disorder differed significantly in terms of the number of bipolar cases managed (*p* = 0.046). Moreover, participants’ attitude scores for SUD, bipolar disorder and schizophrenia varied significantly in terms of their level of confidence in managing the respective cases (see Table 2). A summary of the responses for the assessment of participants’ attitudes toward patients with the three different conditions is presented in Appendix A.

The mean VIF for adjusted linear regression models 1, 2 and 3 was 1.93 (Min VIF = 1.42, Max VIF = 2.93), 1.81 (Min VIF = 1.06, Max VIF = 3.13) and 2.15 (Min VIF = 1.09, Max VIF = 3.21), respectively. According to the multiple linear regression analysis, participants aged ≥27 years had a more positive attitude toward patients with SUD (β = 3.57, 95% CI: 0.39 to 6.75, *p* = 0.018), bipolar disorder (β = 2.74, 95% CI: 0.78 to 3.67, *p* = 0.009) and schizophrenia (β= 1.87, 95% CI: 0.98 to 4.45, *p* = 0.039) compared to their counterparts. Senior psychiatry residents were more likely to have a positive attitude toward patients with SUD (β = 1.71, 95% CI: 0.94 to 4.76, *p* = 0.022), bipolar disorder (β = 2.73, 95% CI: 1.92 to 6.20, *p* = 0.008) and schizophrenia (β = 1.64, 95% CI: 0.90 to 5.21, *p* = 0.032) compared to junior residents. Furthermore, participants who were somewhat confident or very confident in managing SUD, bipolar disorder and schizophrenia patients had a more positive attitude toward patients with the respective conditions (Table 3).

## 4. Discussion

Our study found lower mean attitude scores for psychiatry residents (i.e., negative attitudes or stigmatizing attitudes) toward patients with SUD (mean score: 41.59) than for those with bipolar disorder (mean score: 54.53) and schizophrenia (mean score: 54.53). This finding is supported by a previous investigation [30]. The literature shows that health professionals may have more negative attitudes toward patients with SUD than patients with other physical and mental health conditions. For instance, a European multi-center study showed that health professionals (such as physicians, psychiatrists, psychologists, nurses and social workers) had lower regard for individuals who used substances than for those who were diagnosed with depression or diabetes [29]. Any negative sentiments held by psychiatry residents toward individuals with SUD are alarming, because this negative regard could discourage these patients from seeking help and medication [26,36,37]. In order for patients to be rehabilitated, it is imperative that psychiatry residents adopt the same optimistic attitude toward those who have been diagnosed with drug addiction as they do toward other mental health cases. Limitations in therapeutic options and a lack of experience in psychiatry may affect residents’ attitudes toward patients with different conditions.

Although psychiatry residencies increasingly teach or instruct residents on how to identify and treat individuals with SUD, more emphasis has to be placed on educating residents about common attitudes toward these patients. Psychiatric residency programs should begin with fundamental educational didactics and reflection exercises (including discussion time, journaling and mandatory papers) on attitudes toward people who have SUD [28]. These programs should try to create a supportive “hidden curriculum” in their institutions [28]. Psychiatry residents should follow the appropriate strategies to care effectively for individuals with SUD. For example, psychiatry residents should keep in mind that a patient with SUD may frequently feel shame as their predominant emotion during a professional visit [28]. Thus, residents’ positive attitudes and respectfulness are the key elements in managing patients with SUD. Moreover, greater exposure to patients in recovery and more mentoring from senior practitioners are two other possible strategies to improve psychiatric residents’ attitudes [27,28]. In addition, residents’ attitudes toward individuals with SUD can be improved by online training modules on stigma [38].

Our adjusted regression model demonstrated that senior residents and older age (i.e., ≥27 years) were significantly associated with psychiatry residents’ more positive attitudes toward patients with the three conditions. These findings can be explained by the possibility that psychiatry residents’ optimistic outlook is a result of their greater learning and experience in the residency period. Furthermore, with increasing levels of residency, psychiatry residents may be able to provide comprehensive psychiatric services in a culturally responsive manner, which may make them more positive in dealing with addiction and severe mental illness cases. However, our finding contrasts those of a previous study, which showed that senior residents had stronger negative attitudes toward individuals with SUD than junior residents [30]. This finding may be justified by the fact that the repetition of unpleasant experiences (such as resourcefulness, depression, fear, etc.) in handling patients with SUD and co-occurring mental health illnesses may play a significant role in the emergence of these negative attitudes [28,32,39]. Further quantitative or follow-up studies are recommended to gain a deeper understanding of how psychiatry residents′ attitudes toward patients with SUD and mental health disorders are affected by their residency levels or training periods.

Furthermore, we found that psychiatry residents’ confidence levels (somewhat to very) in managing SUD, bipolar disorder and schizophrenia positively influenced their attitudes toward patients with these conditions. There is evidence of an association between residents′ experience and their confidence in managing diseases [40]. This finding can be explained by the general perception that confidence is expected to play an important role in how a medical professional or student makes decisions, uses their skills and interacts with patients [41].

Several limitations should be kept in mind when interpreting the study findings. First, the cross-sectional design limits the study in drawing causal interferences. Second, respondents’ response biases, such as self-report bias (due to self-reported measures), may exist. Third, since participants reported multiple outcomes on a single questionnaire, inflated type I error may occur. Finally, social desirability bias from the respondents (i.e., a tendency to report more socially desirable attitudes and behaviors, hiding their genuine information) may also occur. Notwithstanding its shortcomings, this study has several strong aspects. This is one of the very first studies that has explored psychiatry residents’ attitudes toward patients with SUD and mental health conditions in Saudi Arabia. Being the first investigation in the country, the findings provide baseline data that could assist policymakers and researchers in performing future robust investigations throughout the country. Moreover, the inclusion of almost all psychiatry residents in the country, which resulted in a high response rate (99%), represents an additional strength of this study. Another potential strength of this study lies in its analytical statistics and the validated scale used to measure the study outcomes.

### Implications for Practice

As negative or stigmatizing attitudes among any medical professionals or psychiatrists toward individuals diagnosed with SUD and psychotic illnesses are a matter of concern, the psychiatry residency period is an ideal starting point for the development of positive attitudes. In this study, we assessed psychiatry residents’ attitudes in Saudi Arabia and found they had negative attitudes toward SUD patients. This finding suggests that educators should continue to enhance curricula to incorporate stigma and behavioral approaches to improve psychiatry residents’ attitudes. Intervention programs or online modules on stigma could be a potential strategy to improve psychiatry residents’ attitudes toward patients with SUD [38]. Another key finding of this study is that senior psychiatry residents had more positive attitudes toward patients with SUD, bipolar disorder and schizophrenia. This finding implies that seeking mentorship from senior residents or other clinicians who are experienced in addiction psychiatry could be another potential strategy to improve psychiatry residents’ attitudes. Furthermore, we recommend conducting an in-depth study in Saudi Arabia to determine whether stigmatizing attitudes exist among other medical professionals or students.

## 5. Conclusions

In conclusion, psychiatry residents had comparably lower attitude scores (i.e., negative attitudes or stigmatizing attitudes) for patients with SUD than for those with bipolar disorder and schizophrenia. Additionally, senior residents, older age (i.e., ≥27 years) and a high confidence level were found to be associated with psychiatry residents’ more positive attitudes toward patients with SUD, bipolar disorder and schizophrenia. Future longitudinal studies are recommended to explore the factors behind psychiatry residents’ negative attitudes toward patients with addictive behaviors and mental illnesses.

## Figures and Tables

**Figure 1 behavsci-13-00642-f001:**
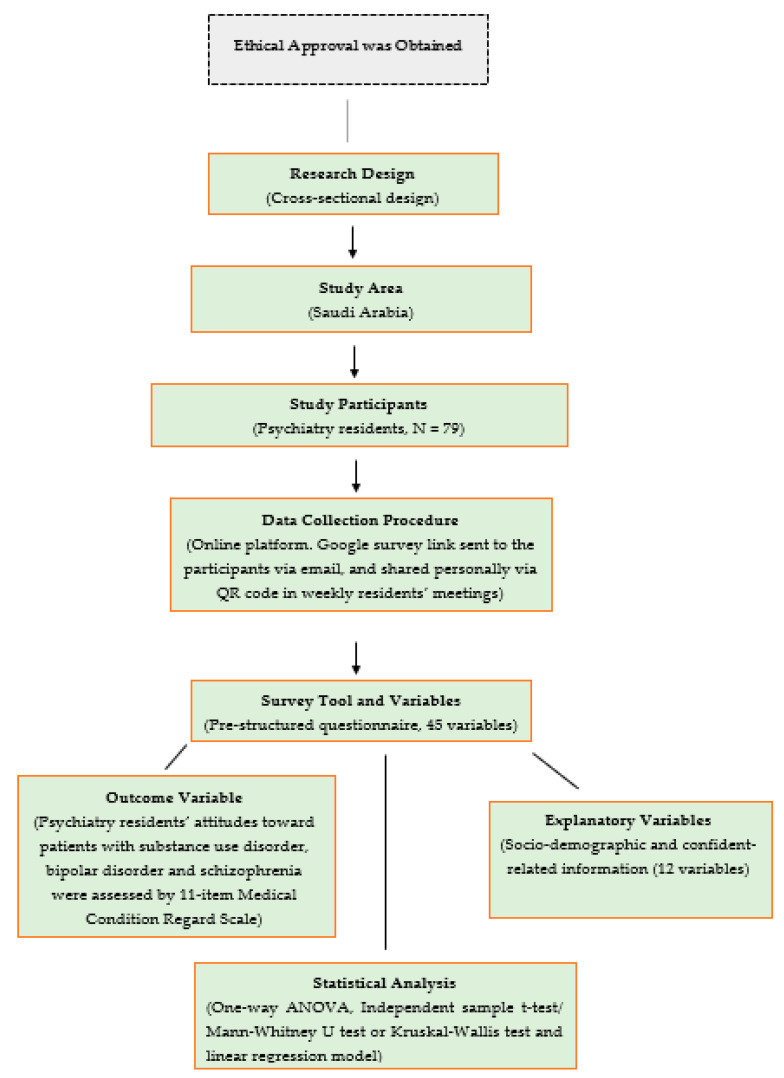
Diagram of study design.

**Table 1 behavsci-13-00642-t001:** Characteristics of the study participants (N = 79).

Variable(s)	Frequency	Percentage (%)
Background information		
Gender		
Male	42	53.2
Female	37	46.8
Age		
<27 years	33	41.8
≥27 years	46	58.2
Marital status		
Single	24	30.4
Married	55	69.6
Level of residency		
Junior	37	46.8
Senior	42	53.2
Smoking status		
Yes	12	15.2
No	67	84.4
Completion of addiction rotation		
Yes	40	50.6
No	39	49.4
Managed addiction cases (substance use)		
≥10 cases	44	55.7
<10 cases	35	44.3
Managed bipolar cases		
≥10 cases	53	67.1
<10 cases	22	32.9
Managed schizophrenia cases		
≥10 cases	56	70.9
<10 cases	23	29.1
How confident are you in managing the following cases?		
Confident in managing addiction cases (substance use)		
Not at all confident	8	10.1
Somewhat confident	42	53.2
Very confident	29	36.7
Confident in managing bipolar cases		
Not at all confident	5	6.3
Somewhat confident	32	40.5
Very confident	42	53.2
Confident in managing schizophrenia cases		
Not at all confident	6	7.6
Somewhat confident	39	49.4
Very confident	34	43.0

**Table 2 behavsci-13-00642-t002:** Participants’ attitudes toward patients with three different conditions by their background characteristics and confidence levels.

Variable(s)	Attitude Score toward Three Conditions					
	Substance Use Disorder †		Bipolar Disorder †		Schizophrenia ††	
	Mean (SD)	*p* Value	Mean (SD)	*p* Value	Mean (SD)	*p* Value
Overall Attitude Score	41.59 (8.09)		54.53 (5.90)		54.20 (6.60)	
Gender		0.256		0.070		0.961
Male	41.95 (7.50)		53.40 (5.93)		53.95 (7.16)	
Female	41.19 (8.79)		55.81 (5.68)		54.48 (5.99)	
Age		**0.011**		**0.004**		**0.009**
<27 years	37.21 (7.02)		50.30 (6.40)		49.15 (7.12)	
≥27 years	42.12 (9.27)		55.70 (5.59)		56.96 (6.17)	
Marital status		0.462		0.297		0.564
Single	41.14 (8.08)		54.09 (6.09)		53.89 (7.05)	
Married	42.63 (8.17)		55.54 (5.41)		54.92 (5.53)	
Level of residency		**0.015**		**0.005**		**0.021**
Junior	39.90 (8.09)		46.57 (6.25)		49.88 (7.33)	
Senior	43.24 (8.17)		54.49 (5.55)		54.58 (5.75)	
Smoking status		0.212		0.349		0.816
Yes	38.91 (7.67)		54.50 (4.95)		55.08 (6.32)	
No	42.07 (8.12)		54.54 (6.08)		54.04 (6.69)	
Completion of addiction rotation		0.220		0.213		0.973
Yes	40.76 (8.71)		54.55 (5.40)		54.45 (5.99)	
No	42.44 (7.42)		54.51 (6.44)		53.95 (7.25)	
Managed addiction (SUD) cases		0.059		0.299		0.327
≥10 cases	42.43 (7.33)		55.15 (5.59)		55.11 (6.28)	
<10 cases	40.93 (8.67)		53.74 (6.25)		53.05 (6.90)	
Managed bipolar cases		0.315		**0.046**		0.638
≥10 cases	41.39 (8.38)		56.01 (5.57)		54.69 (6.20)	
<10 cases	42.00 (7.61)		53.54 (6.52)		53.19 (7.37)	
Managed schizophrenia cases		0.146		0.098		0.520
≥10 cases	41.23 (8.60)		55.30 (5.47)		54.82 (5.88)	
<10 cases	42.48 (6.77)		52.65 (6.59)		52.69 (8.05)	
Confident in managing addiction cases		**<0.001**	NA		NA	
Not at all confident	27.37 (5.28)					
Somewhat confident	38.93 (4.66)					
Very confident	49.38 (3.49)					
Confident in managing bipolar cases				**<0.001**		
Not at all confident			46.40 (6.69)			
Somewhat confident	NA		53.31 (5.08)		NA	
Very confident			56.43 (5.42)			
Confident in managing schizophrenia cases						**<0.001**
Not at all confident					42.16 (5.63)	
Somewhat confident	NA		NA		52.33 (5.29)	
Very confident					58.47 (4.09)	

Note: Bolded values indicate statistically significant results (i.e., *p* < 0.05). † *p* value was estimated by independent-sample *t*-test or one-way ANOVA. †† *p* value was calculated by Mann–Whitney U test or Kruskal–Wallis test. NA = not applicable (i.e., mean difference was not calculated).

**Table 3 behavsci-13-00642-t003:** Multiple linear regression showing the factors associated with participants’ attitudes towards substance use disorder, bipolar disorder and schizophrenia patients.

Variable(s)	Attitude Score toward Three Conditions											
	Regression Model 1: Substance Use Disorder				Regression Model 2: Bipolar Disorder				Regression Model 2: Schizophrenia			
	β	95% CI	*p*	VIF	β	95% CI	*p*	VIF	β	95% CI	*p*	VIF
Gender				1.42				1.73				1.63
Male	Ref.				Ref.				Ref.			
Female	−1.62	−5.78, 2.53	0.438		1.98	−0.93, 4.91	0.179		0.58	−2.78, 3.94	0.733	
Age				1.59				1.44				1.71
<27 years	Ref.				Ref.				Ref.			
≥27 years	3.57	0.39, 6.75	**0.018**		2.74	0.78, 3.67	**0.009**		1.82	0.98, 4.45	**0.039**	
Marital status				1.61				1.22				1.09
Single	Ref.				Ref.				Ref.			
Married	1.69	−2.78, 6.17	0.452		1.21	−1.93, 4.36	0.443		0.43	−3.19, 4.05	0.814	
Level of residency				1.82				1.06				2.13
Junior	Ref.				Ref.				Ref.			
Senior	1.71	0.94, 4.76	**0.022**		2.73	1.92, 6.20	**0.008**		1.64	0.90, 5.21	**0.032**	
Smoking status				2.11				1.24				2.05
Yes	Ref.				Ref.				Ref.			
No	2.83	−2.90, 8.58	0.328		−0.39	−4.42, 3.64	0.847		−0.39	−5.03, 4.24	0.865	
Completion of addiction rotation				2.93				3.13				3.21
Yes	Ref.				Ref.				Ref.			
No	3.76	−5.54, 13.06	0.422		3.19	−3.33. 9.72	0.332		4.14	−3.37, 11.66	0.275	
Managed addiction cases				1.56				2.01				2.33
≥10 cases	−0.48	−6.77, 5.80	0.879		2.35	−2.06, 6.76	0.291		3.38	−1.69, 8.46	0.188	
<10 cases	Ref.				Ref.				Ref.			
Managed bipolar cases				2.31				1.37				1.94
≥10 cases	1.13	−6.90, 9.17	0.780		2.41	−8.06, 3.22	0.195		1.13	−5.64, 5.35	0.728	
<10 cases	Ref.				Ref.				Ref.			
Managed schizophrenia cases				2.04				2.27				2.95
≥10 cases	−0.49	−9.31, 8.32	0.911		4.88	−1.30, 9.06	0.120		2.39	−4.73, 9.51	0.506	
<10 cases	Ref.				Ref.				Ref.			
Confident in managing addiction cases				1.88		NI			NI			
Not at all confident	Ref.											
Somewhat confident	3. 77	1.71, 7.11	**<0.001**									
Very confident	4.90	2.11, 9.12	**<0.001**									
Confident in managing bipolar cases								2.64				
Not at all confident					Ref.							
Somewhat confident		NI			5.00	1.11, 9.90	**0.013**			NI		
Very confident					6.11	2.90, 11.7	**<0.001**					
Confident in managing schizophrenia cases												2.46
Not at all confident									Ref.			
Somewhat confident		NI				NI			4.62	1.23, 9.22	**<0.001**	
Very confident									5.10	3.77, 11.11	**<0.001**	

Note: β = regression coefficient, CI = confidence interval, *p* = probability value, VIF = variance inflation factor, Ref. = reference category and NI = not included in regression model. Bolded values indicate statistically significant results (i.e., *p* < 0.05). The R^2^ for adjusted models 1, 2 and 3 was 0.0532, 0.1244 and 0.0724, respectively.

## Data Availability

The data presented in this study are available on reasonable request from the corresponding author.

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
