# Peer review of "Exploration of Psychiatry Residents’ Attitudes toward Patients with Substance Use Disorder, Bipolar Disorder and Schizophrenia in Saudi Arabia"

_behavsci, 2023, doi:10.3390/bs13080642_

Round 1
Reviewer 1 Report
Stigmatizing attitudes toward mental illness is a serious problem that affects therapeutic behavior and treatment outcome. The article makes a comparative observation of the attitude of psychiatric residents to patients with different psychiatric pathologies. The conducted analysis gives us interesting guidelines for analysis when building a therapeutic plan for individual groups of patients.
The abstract is properly structured, consistent, giving a complete overview of the research conducted.
Introduction: It is generally well structured, but some remarks can be made.
There is a lack of contradictions in the literature to present the differences between the analyzed clinical entities, which would appear to be the reason for the differences in the therapists' stigmatizing attitude towards them. It is necessary to analyze the differences and the associated therapeutic resistance in the analyzed mental disorders. It is also the main reason for failure in therapy, long stays in psychiatric wards, comorbidities as well as associated personality traits leading to the development of a certain negative attitude. All of them are also the reason for the formation of the stigmatizing attitude towards them. The references provided for this analysis are few. It is necessary to make a comprehensive analysis of the literature in order to bring out the differences, contradictions and possible consequences of the course of these mental disorders. Their comorbidity, similarities, differences and treatment difficulties. doi: 10.1016/j.drugalcdep.2019.107552. doi: 10.3390/biomedicines10112728. doi: 10.1016/j.conb.2013.01.022. doi: 10.1016/j.drugalcdep.2015.06.043. doi: 10.1007/7854_2021_272. doi: 10.1093/ijnp/pyz064. PMID: 31802122; doi: 10.3389/fpsyt.2022.845493. doi: 10.1016/j.expneurol.2019.01.016 doi: 10.3389/fpsyt.2022.995455.; doi: 10.22365/jpsych.2021.046. PMID: 34990376. doi: 10.26355/eurrev_202007_22281.
References should be increased /at least 50/ in order to convincingly justify the research done.
Materials, methods, clinical contingent
The study design was properly configured. The chosen methods meet the requirements of the study. The statistics are well chosen for the intended purpose.
The results are well presented, clearly illustrated with tables showing the statistical differences between the groups.
The discussion should be changed by discussing the differences in the attitude towards different groups of patients and the more negative attitude towards those with addictions in the context of the clinical course, the clinical picture and the characteristics of patients with addictions compared to those with schizophrenia and bipolar disease. Aren't differences and limitations in therapeutic options for influence the cause of these differences? What is your opinion about the influence of the lack of experience of residents in psychiatry in building an attitude towards patients with different pathologies?
The limitations as well as the possible application in practice are well presented.
The conclusion reflects the article well, but it is necessary to add the possible explanation of these results. This explanation should be inferred from the discussion of the results. Why such differences. This should also be the ""take home massage"" of the article.
The Reviewer
Author Response
Reviewer #1:
Comments and Suggestions for Authors
Stigmatizing attitudes toward mental illness is a serious problem that affects therapeutic behavior and treatment outcome. The article makes a comparative observation of the attitude of psychiatric residents to patients with different psychiatric pathologies. The conducted analysis gives us interesting guidelines for analysis when building a therapeutic plan for individual groups of patients.
- Comments: The abstract is properly structured, consistent, giving a complete overview of the research conducted.
Authors’ response: Thanks for your appreciation.
- Comments: Introduction: It is generally well structured, but some remarks can be made.
There is a lack of contradictions in the literature to present the differences between the analyzed clinical entities, which would appear to be the reason for the differences in the therapists' stigmatizing attitude towards them. It is necessary to analyze the differences and the associated therapeutic resistance in the analyzed mental disorders. It is also the main reason for failure in therapy, long stays in psychiatric wards, comorbidities as well as associated personality traits leading to the development of a certain negative attitude. All of them are also the reason for the formation of the stigmatizing attitude towards them. The references provided for this analysis are few. It is necessary to make a comprehensive analysis of the literature in order to bring out the differences, contradictions and possible consequences of the course of these mental disorders. Their comorbidity, similarities, differences and treatment difficulties. doi: 10.1016/j.drugalcdep.2019.107552. doi: 10.3390/biomedicines10112728. doi: 10.1016/j.conb.2013.01.022. doi: 10.1016/j.drugalcdep.2015.06.043. doi: 10.1007/7854_2021_272. doi: 10.1093/ijnp/pyz064. PMID: 31802122; doi: 10.3389/fpsyt.2022.845493. doi: 10.1016/j.expneurol.2019.01.016 doi: 10.3389/fpsyt.2022.995455.; doi: 10.22365/jpsych.2021.046. PMID: 34990376. doi: 10.26355/eurrev_202007_22281.
Authors’ response: Thanks for your valuable suggestions. We revised the introduction section as per your and other reviewers’ suggestions. Please see the revised manuscript.
- Comments: References should be increased /at least 50/ in order to convincingly justify the research done.
Authors’ response: References are increased and updated.
- Comments: Materials, methods, clinical contingent: The study design was properly configured. The chosen methods meet the requirements of the study. The statistics are well chosen for the intended purpose.
Authors’ response: Thanks for appreciating the study design.
- Comments: The results are well presented, clearly illustrated with tables showing the statistical differences between the groups.
Authors’ response: Thanks for appreciating the result presentation.
- Comments: The discussion should be changed by discussing the differences in the attitude towards different groups of patients and the more negative attitude towards those with addictions in the context of the clinical course, the clinical picture and the characteristics of patients with addictions compared to those with schizophrenia and bipolar disease. Aren't differences and limitations in therapeutic options for influence the cause of these differences? What is your opinion about the influence of the lack of experience of residents in psychiatry in building an attitude towards patients with different pathologies?
Authors’ response: Thanks for your comments. The discussion section is revised as per you and other reviewers’ suggestions. Please see the revised manuscript.
- Comments: The limitations as well as the possible application in practice are well presented.
The conclusion reflects the article well, but it is necessary to add the possible explanation of these results. This explanation should be inferred from the discussion of the results. Why such differences. This should also be the ""take home massage"" of the article.
Authors’ response: Thanks for appreciating the limitation and implication sections.
Reviewer 2 Report
The study and the topic are of interest. The methodology is globally sound, and results, discussion and conclusion are coherent. I have however a few comments which I hope will improve the paper.
Abstract:
Line 30-31: The sentence “The MCRS is a six-point Likert scale that extends from 30strongly disagree (1) to strongly agree (6), with a higher score indicating a positive attitude.” is not necessary in abstract.
Line 38-41: The sentence “educational and practical interventions are recommended to explore the factors” is not appropriate. Maybe it could be better if it was reconstructed.
Introduction:
Line 47-72: The logic of the introduction is not clear. It would be better to highlight the reasons you focus on patients with Substance Use Disorders, Bipolar Disorders and Schizophrenia in Saudi Arabia, such as high prevalence, heavy burden, and poor prognosis.
Line 64-71: It would be better to focus on psychiatry residents rather than general population when introducing the importance of attitudes toward psychiatric disorders.
Materials and Methods & Results:
In the sections 2.4, 3.2 as well as in Table 3, could the authors add also the Tolerance and VIF (Variance Inflation Factor) in the multiple linear regression?
Discussion:
Without supporting by statistical analysis, the conclusion “Our study found that the attitude score of psychiatry residents was relatively lower for patients with SUD than for those with bipolar disorder and schizophrenia” is not appropriate. Authors should include group comparisons of attitude score of psychiatry residents toward patients with SUDs, bipolar disorders, and schizophrenia.
Conclusion:
The sentence “Future longitudinal studies, which may include educational and practical interventions and comparison groups, are recommended to explore the factors behind psychiatry residents’ negative attitude toward patients with addictive behaviors and mental health illnesses” is not appropriate as mentioned above. The aim of interventions is not to explore influencing factors. Authors should reconstruct this sentence.
Author Response
Reviewer #2:
Comments and Suggestions for Authors
The study and the topic are of interest. The methodology is globally sound, and results, discussion and conclusion are coherent. I have however a few comments which I hope will improve the paper.
- a) Abstract:
- Comments: Line 30-31: The sentence “The MCRS is a six-point Likert scale that extends from 30strongly disagree (1) to strongly agree (6), with a higher score indicating a positive attitude.” is not necessary in abstract.
Authors’ response: The sentence is removed from the abstract.
- Comments: Line 38-41: The sentence “educational and practical interventions are recommended to explore the factors” is not appropriate. Maybe it could be better if it was reconstructed.
Authors’ response: Thanks for your nice observation. The sentence is revised as follows:
“Future longitudinal studies are recommended to explore the factors behind psychiatry residents’ negative attitude toward patients with addictive behaviors and mental health illnesses.”
- b) Introduction:
- Comments: Line 47-72: The logic of the introduction is not clear. It would be better to highlight the reasons you focus on patients with Substance Use Disorders, Bipolar Disorders and Schizophrenia in Saudi Arabia, such as high prevalence, heavy burden, and poor prognosis.
Authors’ response: Thanks for your insightful comments. Introduction is revised as per your suggestions. Please have a look at the revised manuscript.
- Comments: Line 64-71: It would be better to focus on psychiatry residents rather than general population when introducing the importance of attitudes toward psychiatric disorders.
Authors’ response: Thank you for this comment. The introduction section is revised and improved as per reviewers’ suggestions. We are optimistic that readers find a logical flow of this introduction.
Materials and Methods & Results:
- Comments: In the sections 2.4, 3.2 as well as in Table 3, could the authors add also the Tolerance and VIF (Variance Inflation Factor) in the multiple linear regression?
Authors’ response: Thanks for this comments. VIF is added in the table 3 and 2.4 section. Please see the revised manuscript.
Discussion:
- Comments: Without supporting by statistical analysis, the conclusion “Our study found that the attitude score of psychiatry residents was relatively lower for patients with SUD than for those with bipolar disorder and schizophrenia” is not appropriate. Authors should include group comparisons of attitude score of psychiatry residents toward patients with SUDs, bipolar disorders, and schizophrenia.
Authors’ response: Thanks for your nice observation. We have re-written the statement, cut down the tone of the sentence. We discussed based of the descriptive findings of attitude score (higher mean attitude score indicates less stigmatize attitude). Please see the revised manuscript.
Conclusion:
- Comments: The sentence “Future longitudinal studies, which may include educational and practical interventions and comparison groups, are recommended to explore the factors behind psychiatry residents’ negative attitude toward patients with addictive behaviors and mental health illnesses” is not appropriate as mentioned above. The aim of interventions is not to explore influencing factors. Authors should reconstruct this sentence.
Authors’ response: Thanks for noticing. The sentence is corrected.
Reviewer 3 Report
Thanks for inviting this paper. Generally, this paper has been well wrapped up. However, I may have some concerns enlisted below.
1. Did the authors statistically test the difference in attitude scores among the three diagnosis groups before claiming lower attitude scores for patients with SUD than those with bipolar disorder and schizophrenia?
2. Given these participants reported multiple outcomes on a single questionnaire, the results should adjust for multiple testing to prevent inflated type I error.
3. There is evidence of the association between residents' experience and confidence in managing diseases (Tsai, M. C., Chou, Y. Y., & Lin, S. J. (2011). Assessment of experience and training needs in adolescent medicine: perspectives from pediatricians. Tzu Chi Medical Journal, 23(2), 37-41). When these factors are entered into a regression model, colinearity among the independent variables should be checked beforehand.
Author Response
Reviewer #3:
Comments and Suggestions for Authors
Thanks for inviting this paper. Generally, this paper has been well wrapped up. However, I may have some concerns enlisted below.
- Did the authors statistically test the difference in attitude scores among the three diagnosis groups before claiming lower attitude scores for patients with SUD than those with bipolar disorder and schizophrenia?
Authors’ response: Thanks for your nice observation. We discussed it based of the descriptive findings of attitude score (higher mean attitude score indicates less stigmatize attitude). As we didn’t do any comparison analysis, we have re-written the statement, cut down the tone of the sentence. Please see the revised manuscript.
- Given these participants reported multiple outcomes on a single questionnaire, the results should adjust for multiple testing to prevent inflated type I error.
Authors’ response: Thanks for your thought-provoking comments. We fitted an adjusted linear model for identifying the factors. Also the possibility of “inflated type I error” is acknowledged in the limitation section.
- There is evidence of the association between residents' experience and confidence in managing diseases (Tsai, M. C., Chou, Y. Y., & Lin, S. J. (2011). Assessment of experience and training needs in adolescent medicine: perspectives from pediatricians. Tzu Chi Medical Journal, 23(2), 37-41). When these factors are entered into a regression model, colinearity among the independent variables should be checked beforehand.
Authors’ response: The sentence is added in the discussion section. Multicolinearity was checked and mentioned.
Reviewer 4 Report
A well-designed and written manuscript that investigated and summarized psychiatry residents’ attitude towards patients with different psychiatric conditions.
1. Add a flowchart or fish bone diagram for this study design.
2. Authors mentioned that estimated 80 psychiatry residents, who are currently working in Saudi Arabia, are invited to participate in this study. Please confirm that all these psychiatric residents are Saudi nationals. In case your study population contains residents from other nationalities then authors must add that characteristic in table 1.
3. The authors must justify why data presented in SD not in SEM? Is their intention being to just measure variability in the population mean?
4. Authors must update the discussion part with steps to be taken to improve residents’ attitudes toward individuals with SUDs.
5. Do authors remove any outliers from their whole data population in this study? If so, what are the criteria applied?
6. Authors must add regression plots for significant variables.
7. The general rule of thumb is that Cronbach’s alpha of .7 and above is good, .8 and above is better, and .9 and above is best. In this study internal consistency for SUD, bipolar disorder and schizophrenia is less than 0.8, that is good. But when we are assessing an attitude of doctors (residents) its better to have a best (>0.8) internal consistency.
Author Response
Reviewer #4:
Comments and Suggestions for Authors
A well-designed and written manuscript that investigated and summarized psychiatry residents’ attitude towards patients with different psychiatric conditions.
- Add a flowchart or fish bone diagram for this study design.
Authors’ response: Added.
- Authors mentioned that estimated 80 psychiatry residents, who are currently working in Saudi Arabia, are invited to participate in this study. Please confirm that all these psychiatric residents are Saudi nationals. In case your study population contains residents from other nationalities then authors must add that characteristic in table 1.
Authors’ response: All these psychiatric residents are Saudi nationals. Non-saudi nationals are not permitted to this residency program.
- The authors must justify why data presented in SD not in SEM? Is their intention being to just measure variability in the population mean?
Authors’ response: Yes, we measured variability in the population mean.
- Authors must update the discussion part with steps to be taken to improve residents’ attitudes toward individuals with SUDs.
Authors’ response: The discussion part is updated. Please read the revised manuscript. Thank you.
- Do authors remove any outliers from their whole data population in this study? If so, what are the criteria applied?
Authors’ response: Data had no outliers. In addition, collinearity was checked by VIF and mentioned in the table. Please see the revised manuscript.
- Authors must add regression plots for significant variables.
Authors’ response: Thanks for your nice suggestion. We, the authors, are discussed about this regression plots. We wouldn’t like to add this plots as the table clearly presents all the information for the readers. We think the addition of regression plot doesn’t add anything new. Please consider this.
- The general rule of thumb is that Cronbach’s alpha of .7 and above is good, .8 and above is better, and .9 and above is best. In this study internal consistency for SUD, bipolar disorder and schizophrenia is less than 0.8, that is good. But when we are assessing an attitude of doctors (residents) its better to have a best (>0.8) internal consistency.
Authors’ response: Thanks for your comments.
Round 2
Reviewer 1 Report
The article has been revised according to the requirements. I believe that in this form it can be submitted for publication.
The Reviewer
Reviewer 3 Report
The authors have addressed my concerns in an indirect way, although the authors did not conduct additional analyses as I suggested. I have no more questions.